# CRL-Net: Accelerated Magnetic Resonance Imaging Reconstruction through Coil Representation Learning

## Abstract

Magnetic Resonance Imaging(MRI) is a lengthy medical scan that stems from a long acquisition time. Its length is mainly due to the traditional sampling theorem, which defines a lower bound for sampling. However, it is still possible to accelerate the scan by using a different approach such as Compress Sensing(CS) or multi-coil Parallel Imaging(PI). These two complementary methods can be combined to achieve a faster scan with high-fidelity imaging. Recent advancements in Deep Learning (DL) have shown the potential to outperform traditional CS reconstruction techniques. This paper introduces CRL-Net, a novel Coil Representation Learning Network for accelerated multi-coil MRI reconstruction. The architecture of CRL-Net comprises a coil-wise encoder, devised to ascertain the distinctive representations of each coil. This is further complemented by a coil-attention layer, which synergistically assimilates inputs from both the sensitivity map estimations and the coil-wise encoder. Comprehensive evaluations of the CRL-Net, using the FastMRI benchmark for multi-coil datasets across knee and brain regions at both 4x and 8x acceleration, manifest significant advancements over the prevailing state-of-the-art methodologies. Such results elucidate the promising capability of CRL-Net in refining the accuracy and efficiency of MRI reconstructions.

## 1 Introduction

Magnetic Resonance Imaging (MRI) is a non-invasive imaging technique used to visualize internal organs without exposing patients to ionizing radiation. To accelerate the acquisition process, two methods have been developed: Parallel imaging (PI) and compressive sensing (CS). PI is a technique that reduces time-consuming phase-encoding steps by utilizing knowledge of the receiver coil placement and its sensitivity to create a special localization signal Pruessmann (2006); Larkman & Nunes (2007). PI produces a complex matrix for each input coil, which is multiplied by the position-dependent coil sensitivity map Katscher & Börnert (2006). Those maps are usually generated from the auto-calibration signal (ACS) that corresponds to the low spatial frequencies Ying & Sheng (2007). However, CS doesn't use complimentary information. CS suggests that signals can be recovered with fewer samples than the traditional sampling theorem requires Candès & Wakin (2008); Yang et al. (2016); Eldar & Kutyniok (2012). In order to apply CS successfully, the signal must be sparse under a known transform domain. Additionally, the sampling must be incoherent to eliminate artifacts such as the aliasing artifact. Moreover, non-linear algorithms should be applied to reconstruct the signal, as suggested by Boyd et al. (2011); Beck & Teboulle (2009); Lustig et al. (2007).

Deep learning (DL) models have shown a remarkable ability to outperform classic CS reconstruction techniques Zhu et al. (2018); Hammernik et al. (2018). Classic CS techniques rely on sparsity-based priors, which are often insufficient to model the complexity of the image data Lustig et al. (2007). Conversely, DL models utilize extensive training datasets to extract more sophisticated, data-specific priors LeCun et al. (2015). These models are capable of learning complex patterns and features in the data, which can lead to improved image quality and finer detail in the reconstructed images Gong et al. (2018).

Recently, numerous studies have employed unrolled methods in the context of MRI reconstruction, often involving a series of linear operations. These operations typically include a reduce operator (such as the inverse Fourier transform and root-sum-of-squares) and an expand operator (which multiplies the image by a sensitivity map estimation and applies a Fourier transform). Furthermore, non-linearity is present due to the Convolutional Neural Network (CNN) method, which serves as a reconstruction network Sriram et al. (2020); Fabian et al. (2023). Despite the complexity of this series of operations, it doesn't necessarily hold more information than a CNN that has learned the linear transformations involved in the linear operators Goodfellow et al. (2016). The reason lies in the inherent structure of CNNs and their ability to learn transformations. CNNs are highly adept at learning both linear and non-linear transformations from the training data LeCun et al. (2015). When a CNN learns the transformations associated with the reduce and expand operators, it is effectively learning to perform these operations in a way that is optimized for the task at hand, in this case, MRI reconstruction Zhang et al. (2017); Cahana et al. (2023). Furthermore, the unrolled methods involve a fixed set of operations that are applied in a particular order. In contrast, a CNN has the flexibility to learn a direct mapping from the input to the output, which can involve complex combinations of transformations that aren't limited to the sequence of reduce, CNN, and expand operations. This direct mapping can potentially capture more intricate relationships in the data, leading to superior performance Goodfellow et al. (2016); Schmidhuber (2015). Thus, while the unrolled methods certainly involve sophisticated operations and can achieve excellent results, they don't inherently hold more information or capability than a well-trained CNN that has learned the transformations associated with these operations LeCun et al. (2015).

In this work, we introduce CRL-Net, a Coil Representation Learning Network designed for multi-coil MRI reconstruction. The primary contributions of this work are as follows:

- We have designed and presented the innovative CRL-Net, a Coil Representation Learning Network specifically tailored for the optimization of multi-coil MRI reconstruction.
- The CRL-Net incorporates a coil-wise encoder, which is a unique architectural component devised to ascertain the distinctive representations of each coil.
- Our model features a coil-attention layer, a novel approach that synergistically assimilates inputs from both the sensitivity map estimations and the coil-wise encoder, thereby enhancing the extraction of pertinent features.
- We conducted exhaustive evaluations of the CRL-Net using the reputable FastMRI benchmark. These evaluations spanned multi-coil datasets across both knee and brain regions, considering both 4x and 8x acceleration parameters.

Our findings reveal that the CRL-Net offers significant advancements over current state-of-the-art methodologies, underscoring its potential as a leading tool in the MRI reconstruction domain.

## 2 RELATED WORK

Deep learning has shown significant potential across a variety of machine learning problems, and this extends to the domain of computational MRI. In this section, we briefly introduce the most recent developments in accelerated MRI techniques. Later, in section results we will draw quantative comparisons between these methodologies and our proposed method. The invertible variant of Recurrent Inference Machines (i-RIM) Putzky et al. (2019) uses an iterative map that combines the current reconstruction, a concealed memory state, and the likelihood gradient. This model encapsulates vital information about the recognized generative process and evaluates the fidelity of the measurements it reproduces. The XPDNet Ramzi et al. (2020) is an end-to-end unrolled cross-domain network model based on the Chambolle-Pock algorithm. This model leverages the final five unrolled iterations to ensure the learning of a complex non-linear acceleration scheme. The Σ-net Hammernik et al. (2019) is built on the analysis of contemporary network designs, such as DUNs, U-net, and various data consistency terms. The Σ-net employs strategies involving adversarial losses and self-supervised fine-tuning to enhance image quality. Uniquely, it utilizes a sensitivity-weighted coil-combination reference, which minimizes noise-induced bias and allows for more equitable comparisons. The End-to-End Variational Network (E2E-VarNet) Sriram et al. (2020), inspired by the Variational Network for MRI reconstruction Hammernik et al. (2018), employs an unroll-based method. Each cascade in this model contains a reduce operator, data consistency, and an expand

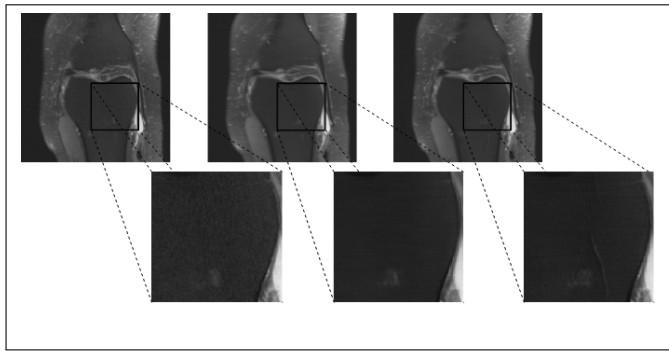

Figure 1: Qualitative results on knee sample. left to right: target, CRL-Net, E2E-VarNet

operator which expands the image based on the coil sensitivity estimator. The HUMUS-Net Fabian et al. (2023), employs convolutional blocks for the extraction of high-resolution features and utilizes a Transformer-based extractor for the refinement of low-resolution features. This process leads to the generation of a high-resolution output. It is noteworthy to mention that the evaluation of HUMUS-Net was exclusively conducted on the fastMRI knee dataset

## 3 METHOD

Our proposed method, the CRL-Net, offers a pioneering approach to MRI reconstruction, seamlessly combining sensitivity map estimation with a Coil Representation Learning Network. By harnessing parallel imaging techniques and employing a multi-coil architecture complemented by a coil-attention mechanism, the CRL-Net ensures superior image reconstruction[Figure 2].

### 3.1 SENSITIVITY MAP ESTIMATOR

Parallel imaging techniques involve acquiring data from multiple receiver coils simultaneously to expedite the imaging process. However, as the sensitivity of each coil to the signal varies across the imaging field, accurate estimation of the sensitivity maps for each coil is crucial to account for these variations and ensure proper data combination during image reconstruction Larkman & Nunes (2007); Sriram et al. (2020).

The sensitivity maps can be estimated in advance through various methods Pruessmann et al. (1999); Uecker et al. (2014), or acquired through end-to-end learning via a SME network as proposed in Sriram et al. (2020). In this study, our objective is to develop a holistic approach, and therefore we employ the latter technique to estimate the sensitivity maps. Specifically, we estimate the sensitivity maps from the undersampled input measurements' low-frequency, equivalent to the auto-calibrationl-signals, region during the training phase using a conventional U-Net network.

### 3.2 COIL REPRESENTATION LEARNING NETWORK

Our primary focus in this study is to develop an advanced coil-centric method adept at handling multi-coil MRI input. In the subsequent section, we detail each component of our method. In addition to the following methods, we employ the common practice of Fourier transform to map the frequency domain to the image domain. Furthermore, when using the brain dataset, we apply coil standardization Yiasemis et al. (2022).

#### 3.2.1 ENCODER DESIGN

Our encoder comprises four Coil-Wise Blocks(CWB), each characterized by depthwise convolutions. Each block amalgamates depthwise convolution operations, group normalization, and employs the Leaky Rectified Linear Unit (ReLU) as its activation function. By deploying depthwise convolutions, we aim to extract features unique to each coil. This approach is further augmented by the introduction of group normalization, ensuring the model processes multi-coil data optimally.

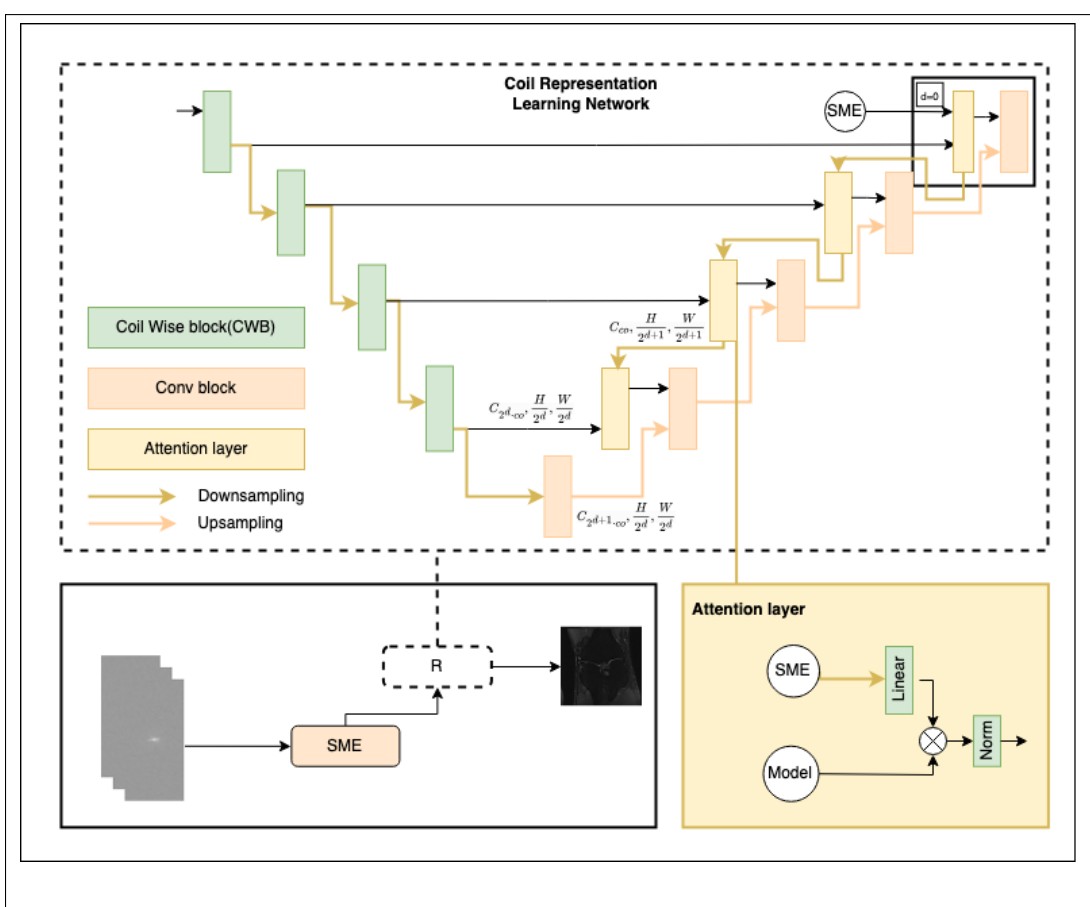

Figure 2: Bottom-left: The proposed CRL-Net consists of two main parts: Sensitivity Map Estimator, and Reconstruction model. Top: The CRL-Net is displayed, for readability, we retain only the mathematical formula inputs of one decoder layer. Bottom-right: The Coil-Attention mechanism is detailed, starting with a downsampling process followed by a linear transformation, then coil-wise multiplication, and finally a softmax operation for normalization.

### 3.2.2 DECODER DESIGN

In contrast, our decoder is structured around four convolution blocks. Each of these integrates a convolution operation, instance normalization, and a Leaky ReLU activation function. We've significantly enhanced our model by integrating a coil-attention mechanism, activated before each convolution block. This mechanism works in tandem with the parallel encoder feature and the SME output.

The cail-attention mechanism's core idea is to highlight pivotal features. This is executed by downsampling the SME results, projecting them onto the model's feature space, multiplying them with the parallel encoder output, and finally, normalizing these results using a softmax operation. This sequence ensures the emphasis of essential features during the reconstruction process.

Specifically, let $d \in \{0, ..., D\}$ where $D$ denotes the total decoder layers, and $co \in \mathcal{N}$ represents the number of coils. Every layer accepts three inputs:

1. The matching encoder layer with dimensions $(\mathcal{C}_{2^d \times co}, \frac{H}{2^d}, \frac{W}{2^d})$.

2. The SME results with dimensions $(Cco, \frac{H}{2^{d+1}}, \frac{W}{2^{d+1}})$, post the application of the downsampling operation (DS) at the $d+1$ stage.

3. The output from the prior decoder layer with dimensions $(\mathcal{C}_{2^{d+1} \times co}, \frac{H}{2^d}, \frac{W}{2^d})$.

First, the DS operation refines the SME results, mapping $S \in \mathbb{R}^{\frac{H}{2^{d+1}} \times \frac{W}{2^{d+1}}} \rightarrow \mathbb{R}^{\frac{H}{2^d} \times \frac{W}{2^d}}$. Following this, a linear transformation adjusts coil weights. The results are then coil-wise multiplied with the encoder's parallel output, followed by normalization using a softmax operation. Mathematically, this is:

$$Coil - Attention(CWB, S) = Softmax(CoilMul(Linear(DS(S)), CWB)) \qquad (1)$$

After determining the coil-attention output, it's utilized by the decoder to compute the output for layer $d$. This output is concatenated with the output from the preceding layer $Decoder_{d-1}$ and then directed through a convolutional block (CB). The CB comprises two convolution layers, instance normalization, and a Leaky ReLU. The ultimate layer of the decoder produces the fully reconstructed image, which is mathematically represented as:

$$Decoder_d = CB(Con(Coil - Attention_d(CWB_d, S_{d+1}), Decoder_{d-1})) \qquad (2)$$

Efficiently, $D$ downsamples of the SME results are precomputed and sequentially fed into the appropriate coil-attention layer $d$.

A notable point is the top-bottom mapping of the decoder layers, where layer 0 corresponds to the final decoder layer. This convention is chosen to simplify mathematical notation.

### 3.3 LOSS FUNCTION

The Structural Similarity Index Measurement (SSIM) and L1, also referred to as the mean absolute error, are two commonly employed metrics for assessing the quality of image reconstructions in MRI. The key objective of both SSIM and L1 is to evaluate the similarity between the reconstructed and original images and minimize the discrepancy between them. SSIM is a full-reference metric that quantifies the similarity between two images based on their luminance, contrast, and structure. In contrast, L1 is a straightforward, pixel-wise metric that computes the absolute difference between each pixel in the reconstructed image and the original image. A combination of SSIM and L1 can offer a more thorough evaluation of the quality of an MRI reconstruction, as SSIM captures structural details while L1 measures pixel-level disparities. An enhanced version of SSIM, known as Multiscale SSIM (MS-SSIM), is implemented over multiple scales using a multi-step downsampling method. This process mirrors the multiscale processing characteristic of the early visual system. MS-SSIM has been shown to perform as well as or even better than SSIM across a variety of subjective image and video databases. Consequently, our loss function comprises a weighted combination of L1 and MS-SSIM.

$$\mathcal{L} = \alpha * L_1(x, y) + (1 - \alpha) * (1 - MS\text{-}SSIM(x, y)) \tag{3}$$

## 4 EXPERIMENTS

In the following section, we will delve into the details of the FastMRI benchmark and the metrics employed for evaluation

### 4.1 FASTMRI BENCHMARK

FastMRI is the largest benchmark for accelerated MRI scans. The benchmark is divided into training, validation, and test subsets. These subsets contain k-space data, a complex-valued matrix that represents MR images in frequency space. This data has been subjected to undersampling masks to mimic the real-world process of MRI data acquisition.

FastMRI includes data from two types of scans: knee and brain. For knee scans, multi-coil raw data was gathered from 1,594 diagnostic MRI scans using various clinical MRI systems. On the other hand, the brain data came from 6,970 fully sampled brain MRIs, collected using a range of MRI machines across multiple clinical locations.

The ground truth for the multi-coil dataset is provided by applying the root-sum-of-squares reconstruction method to the fully sampled k-space data. All ground truth images are cropped to the central $320 \times 320$ pixel region.

### 4.2 EVALUATION METRICS

To evaluate our method, we adopted three prominent metrics from the literature on image reconstruction: Structural Similarity Index Measure (SSIM) Wang et al. (2004), Peak Signal-to-Noise Ratio (PSNR), and Normalized Mean Square Error (NMSE).

- **Structural Similarity Index Measure (SSIM)**: SSIM endeavors to gauge the perceived quality of an image. It is formally defined by:

$$SSIM(x, y) = l(x, y)^\alpha \times c(x, y)^\beta \times s(x, y)^\gamma \tag{4}$$

  where $\alpha$, $\beta$, and $\gamma$ are constants. For validation, we employed the E2E-VarNet method, using a window size of $7 \times 7$, parameters $k_1 = 0.001$ and $k_2 = 0.003$, and $L = \max(v)$, where $v$ denotes the maximum value of the target volume Nilsson & Akenine-Möller (2020).

- **Normalized Mean Square Error (NMSE)**: NMSE is a pixel-wise metric that calculates the discrepancy between a reconstructed image $x$ and a target image $y$. It is defined as Chandler (2013):

$$NMSE(x, y) = \frac{||x - y||_2^2}{||y||_2^2} \tag{5}$$

- **Peak Signal-to-Noise Ratio (PSNR)**: Commonly utilized in Compressed Sensing (CS) applications, PSNR assesses the quality of image reconstructions. Its definition is Chandler (2013):

$$PSNR(x, y) = 10 \times \log_{10}\left(\frac{y^2}{MSE(x, y)}\right) \tag{6}$$

## 5 RESULTS

In this section, we present the performance of our proposed CRL-Net on the FastMRI test set with two different acceleration factors (4x and 8x) and compare it with several existing methods. The results are separately discussed for brain scans and knee scans.

Table 1: Quantitative results on FastMRI test set with Acceleration Factor = 4

| Scan | Model | SSIM ($\uparrow$) | NMSE ($\downarrow$) | PSNR($\uparrow$) |
|------|-------|------|------|------|
| Brain | Unet Ronneberger et al. (2015) | 0.946 | 0.007 | 38 |
| | XPD-net Ramzi et al. (2020) | 0.959 | 0.003 | 41 |
| | E2E-VarNet Sriram et al. (2020) | $0.959 \pm$ 2e-3 | $0.004 \pm$ 2e-4 | $41 \pm 0.3$ |
| | **CRL-Net** | **0.970** $\pm$ 3e-3 | **0.003** $\pm$ 3e-4 | **43** $\pm 0.4$ |
| Knee | Unet Ronneberger et al. (2015) | 0.91 | 0.007 | 34 |
| | i-RIM Putzky et al. (2019) | 0.928 | 0.005 | 40 |
| | $\Sigma$-net Hammernik et al. (2019) | 0.928 | 0.005 | 40 |
| | XPD-net Ramzi et al. (2020) | 0.959 | 0.003 | 41 |
| | E2E-VarNet Sriram et al. (2020) | $0.930 \pm$ 2e-3 | $0.005 \pm$ 2e-4 | $40 \pm 0.3$ |
| | **CRL-Net** | **0.934** $\pm$ 2e-3 | **0.004** $\pm$ 5e-4 | **42** $\pm 0.5$ |

## 5.1 4X ACCELERATION FACTOR

The results on the FastMRI test set with an acceleration factor of 4x are shown in Table 1. Our proposed CRL-Net has demonstrated superior performance for both brain and knee scans, outperforming other models such as Unet, XPD-net, and E2E-VarNet.

For brain scans, CRL-Net achieved the highest SSIM of 0.970, NMSE of 0.003, and PSNR of 43. These results are better than the other models, with the closest competitor being the XPD-net and E2E-VarNet that both achieved an SSIM of 0.959 and a PSNR of 41.

Similarly, for knee scans, our CRL-Net achieved an SSIM of 0.934, NMSE of 0.004, and PSNR of 42. Again, these results are superior to the other models, with the closest competitor being the XPD-net with an SSIM of 0.959 and a PSNR of 41.

## 5.2 8X ACCELERATION FACTOR

The results on the FastMRI test set with an acceleration factor of 8x are shown in Table 2. Similar to the 4x acceleration factor, our proposed CRL-Net outperformed the other models for both brain and knee scans.

In the case of brain scans, the CRL-Net achieved the highest SSIM of 0.957, NMSE of 0.004, and PSNR of 42, outperforming the closest competitor, the E2E-VarNet, which achieved an SSIM of 0.943 and a PSNR of 38.

For knee scans, the CRL-Net again achieved the highest performance with an SSIM of 0.898, NMSE of 0.007, and PSNR of 38. The closest competitor was the XPD-net with an SSIM of 0.942 and a PSNR of 38.

In summary, our proposed CRL-Net consistently outperforms other models on the FastMRI test set for both acceleration factors and both types of scans, demonstrating its superior performance in MRI reconstruction.

## 5.3 EVALUATION OF WEIGHTED LOSS IMPACT

We carried out an experiment to ascertain the significance of the weighted loss on the overall outcomes. The results of this experiment are summarized in Table 3.

Referring to Table 3, it is evident that utilizing a weighted sum yields superior results. In addition to evaluating the presence or absence of the weighted sum, we also investigated the most suitable values of the alpha parameter for the weighted loss. Further details are provided in the supplementary material.

Table 2: Quantitative results on FastMRI test set with Acceleration Factor = 8

| Scan | Model | SSIM ($\uparrow$) | NMSE ($\downarrow$) | PSNR($\uparrow$) |
|------|-------|------|------|------|
| Brain | Unet Ronneberger et al. (2015) | 0.922 | 0.014 | 35 |
| | XPD-net Ramzi et al. (2020) | 0.942 | 0.007 | 38 |
| | E2E-VarNet Sriram et al. (2020) | $0.943 \pm$ 5e-3 | $0.007 \pm$ 4e-4 | $38 \pm 0.4$ |
| | **CRL-Net** | $\textbf{0.957} \pm$ 4e-3 | $\textbf{0.004} \pm$ 4e-4 | $\textbf{42} \pm 0.5$ |
| Knee | Unet Ronneberger et al. (2015) | 0.864 | 0.013 | 35 |
| | i-RIM Putzky et al. (2019) | 0.888 | 0.009 | 37 |
| | $\Sigma$-net Hammernik et al. (2019) | 0.888 | 0.009 | 37 |
| | XPD-net Ramzi et al. (2020) | 0.942 | 0.007 | 38 |
| | HUMUS-Net-L Fabian et al. (2023) | 0.8944 | 37.3 | 0.0083 |
| | E2E-VarNet Sriram et al. (2020) | $0.890 \pm$ 2e-3 | $0.009 \pm$ 3e-4 | $37 \pm 0.3$ |
| | **CRL-Net** | $\textbf{0.898} \pm$ 3e-3 | $\textbf{0.007} \pm$ 4e-4 | $\textbf{38} \pm 0.4$ |

Table 3: w/wo weighted loss

| | with | without | SSIM ($\uparrow$) | NMSE ($\downarrow$) | PSNR ($\uparrow$) |
|------|------|------|------|------|------|
| 4x | - | $\checkmark$ | 0.89 | 0.005 | 40 |
| | $\checkmark$ | - | 0.934 | 0.004 | 42 |
| 8x | - | $\checkmark$ | 0.871 | 0.009 | 34 |
| | $\checkmark$ | - | 0.898 | 0.007 | 38 |

# 6 LIMITATION

The method presented in this paper has several limitations that need to be addressed. Firstly, there may exist a gap between the quantitative results presented in this study and the subjective reconstruction quality as evaluated by radiologists. This discrepancy raises important issues that require further exploration and discussion between computer science and radiology professionals.

Secondly, our evaluation is solely limited to the FastMRI benchmark dataset, and we did not evaluate our method on other datasets due to their smaller size compared to the FastMRI dataset.

Lastly, we only evaluated our method at 4x and 8x acceleration factors, although our method is capable of accelerating beyond these factors. We employed these acceleration factors to compare our method's performance against other methods in the literature.

# 7 CONCLUSIONS

Throughout our study, the performance of CRL-Net was meticulously assessed against several existing MRI reconstruction models, specifically on the FastMRI test set. Notably, CRL-Net exhibited consistent superiority across different acceleration factors (both 4x and 8x) and distinct scan types (brain and knee). These findings robustly validate the prowess of the CRL-Net in delivering high-quality MRI reconstructions.

When juxtaposed with other established models such as the Unet, XPD-net, and E2E-VarNet, CRL-Net's prowess became distinctly evident. For instance, under a 4x acceleration factor, CRL-Net not only achieved the highest SSIM, NMSE, and PSNR for both brain and knee scans but also showed a marked improvement over its closest competitors. This trend of excellence was mirrored in the 8x acceleration scenario.

Furthermore, our exploration into the impact of the weighted loss confirmed its significance in enhancing the overall performance. The results highlighted the merits of adopting a weighted sum in the model's architecture, with subsequent evaluations identifying the optimal alpha parameter values for this weighted loss.

In essence, the evidence presented firmly establishes CRL-Net's position as a leading technique in MRI reconstruction. Its combination of sophisticated design and effective use of weighted loss offers promising avenues for future research and practical applications in the realm of medical imaging.

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
