# OpenReview forum: "CRL-NET: ACCELERATED MAGNETIC RESONANCE IMAGING RECONSTRUCTION THROUGH COIL REPRESENTATION LEARNING"
_ICLR.cc/2024/Conference — Submitted to ICLR 2024_

### Official Review · Reviewer_3r6G · 2023-10-23

**Soundness:** 1 poor
**Presentation:** 1 poor
**Contribution:** 1 poor
**Rating:** 1
**Confidence:** 5

**Summary:**

The paper titled: CRL-NET: ACCELERATED MAGNETIC RESONANCE IMAGING RECONSTRUCTION THROUGH COIL REPRESENTATION LEARNING presents a novel network architecture to encode coil information into DL-based multi-coil MRI reconstruction, showing improved quantitative results.

**Strengths:**

1. This paper present an interesting architecture to learn the coil representation for DL-based multi-coil reconstruction, the coil-attention module is inspiring in considering multi-coil information into DL-based reconstruction.

**Weaknesses:**

In general, I don't think this paper meets the standard of ICLR and other related conferences.
1. There is no visual comparisons between the proposed method and other benchmarks, quantitative results can deliver some information, but for medical imaging, or MRI, its important to visualize the reconstruction results, how it deals with hallucinations, but there is no visualization at all.

2. The ablation studies are so weak, this paper proposes coil-attention, and coil representation learning. I was expecting ablation studies on those architecture components, instead of weighted losses, which is a trivial ablation study.

3. Maybe I missed it, seems this work is not based-on unrolled networks. Therefore, I would assume there might be generalizability problems in terms of sampling patterns and others. The authors didn't make any analysis.

4. The paper is poorly written, all the tables are not professional, I haven't seen any paper reports PSNR without decima, and the papers are so redundant, I can barely retrieve any information.

**Questions:**

1. How does the coil-attention module help, why it works?

---

### Official Review · Reviewer_Q3j2 · 2023-10-30

**Soundness:** 1 poor
**Presentation:** 1 poor
**Contribution:** 1 poor
**Rating:** 1
**Confidence:** 4

**Summary:**

The paper presents a technique for rapid parallel MRI reconstruction leveraging deep neural networks, named as the CRL-Net. The CRL-Net is composed of three components: a coil sensitivity map normalization module, an estimation module and an image reconstruction module that integrates both attention and convolutional layers. Experiments on the fastMRI dataset demonstrate the superior performance of the model.

**Strengths:**

1, The paper is overall well motivated, clearly written and accessible to non-MRI experts.

**Weaknesses:**

1, The innovative aspects of the proposed method are quite restrained, as the majority of its components are well-established and extensively documented in existing literature. The joint coil sensitive map estimation and image reconstruction has been already proposed in E2E-VarNet. The attention mechanism is also nowadays widely used for MRI and other medical problems, such as [R1], HUMUS-Net: Hybrid Unrolled Multi-Scale Network Architecture for Accelerated MRI Reconstruction. NeurIPS 2022.

2, The qualitative analysis is poorly done in the paper. While commonly used quantitative metrics like PSNR or SSIM may be essential for research consistency, they often fall short in accurately assessing the quality of medical images. A more robust qualitative comparison would have been beneficial.

3, The paper is not very well-written, and the quality can be improved noticeably.

**Questions:**

1, The authors should provide more qualitative results in Figure 1.

2, The paper spends almost a papge to introduce fMRI dataset and PSNR/SSIM/NMSE metrics. But, the authors only provide very few visual or ablation studies in the main paper.

---

### Official Review · Reviewer_H3Dq · 2023-11-04

**Soundness:** 1 poor
**Presentation:** 1 poor
**Contribution:** 2 fair
**Rating:** 3
**Confidence:** 4

**Summary:**

The authors presented a Coil Representation Learning Network designed for multi-coil MRI reconstruction. The paper needs to

**Strengths:**

The authors have investigated the effects of two distinct acceleration factors, 4X and 8X, on MRI reconstruction quality. Furthermore, they provided evidence that a weighted fusion of SSIM and L1 loss metrics could lead to performance improvements.

**Weaknesses:**

The paper would benefit from a thorough review to address the numerous grammatical errors that currently hinder the reading experience. Additionally, while the integration of SSIM and L1 losses is standard in image reconstruction tasks, it is imperative for the authors to consider expanding upon this normative approach. The reliance on evaluation metrics such as SSIM, NMSE, and PSNR is somewhat restrictive and may not fully capture the diagnostic quality of radiographic imaging. Inclusion of expert assessments from radiologists could significantly enhance the validation of the reconstructed MRI's diagnostic utility, which is paramount in medical applications.

**Questions:**

Could the authors elaborate on their decision to employ SSIM as the loss function rather than MSE loss? An explanation of this choice would provide clarity on the advantages or the intended outcomes of using SSIM over the traditional MSE, particularly in the context of the specific reconstruction challenges this work addresses.

---

### Meta-Review · Area_Chair_QMKL · 2023-12-16

**Metareview:**

The paper presents a Coil Representation Learning Network designed for multi-coil MRI reconstruction. The CRL-Net is composed of three components: a coil sensitivity map normalization module, an estimation module and an image reconstruction module that integrates both attention and convolutional layers. Experiments on the fastMRI dataset demonstrate the superior performance of the model.

All reviewers provided reject rating and authors did not submit a response.

Strengths:
+ The paper is well-motivated and accessible to non-MRI experts.

Weaknesses:
- Novelty is limited: integration of SSIM and L1 losses is standard in image reconstruction tasks
- The qualitative analysis requires additional work
- The paper does not provide comparison with other methods
- Writing quality of the paper can be significantly improved

**Justification For Why Not Higher Score:**

None

**Justification For Why Not Lower Score:**

N/A

---

### Decision · Program_Chairs · 2024-01-16

Reject